# Impact of High-Dose Methotrexate on the Outcome of Patients with Diffuse Large B-Cell Lymphoma and Skeletal Involvement

**DOI:** 10.3390/cancers13122945

**Published:** 2021-06-12

**Authors:** Mélanie Mercier, Corentin Orvain, Laurianne Drieu La Rochelle, Tony Marchand, Christopher Nunes Gomes, Aurélien Giltat, Jérôme Paillassa, Aline Clavert, Jonathan Farhi, Marie-Christine Rousselet, Emmanuel Gyan, Roch Houot, Marie-Pierre Moles-Moreau, Mathilde Hunault-Berger

**Affiliations:** 1Maladies du Sang, CHU d’Angers, 49000 Angers, France; christopher.nunesgomes@chu-angers.fr (C.N.G.); aurelien.giltat@chu-angers.fr (A.G.); jerome.paillassa@chu-angers.fr (J.P.); aline.clavert@chu-angers.fr (A.C.); jonathan.farhi@chu-angers.fr (J.F.); MPMoles@chu-angers.fr (M.-P.M.-M.); mahunault@chu-angers.fr (M.H.-B.); 2Service d’Hématologie, CH Bretagne Atlantique, 56000 Vannes, France; 3Fédération Hospitalo-Universitaire Grand-Ouest Acute Leukemia (FHU-GOAL), 49033 Angers, France; L.DRIEULAROCHELLE@chu-tours.fr (L.D.L.R.); emmanuel.gyan@univ-tours.fr (E.G.); 4Université d’Angers, Inserm, CRCINA, 49000 Angers, France; 5Service d’Hématologie et Thérapie Cellulaire, Centre Hospitalier Universitaire, Université de Tours, 37000 Tours, France; 6Service d’Hématologie Clinique, CHU de Rennes, 35000 Rennes, France; tony.marchand@chu-rennes.fr (T.M.); Roch.HOUOT@chu-rennes.fr (R.H.); 7Département de Pathologie Cellulaire et Tissulaire, CHU d’Angers, 49000 Angers, France; mcrousselet@chu-angers.fr

**Keywords:** lymphoma, bone, skeletal, high-dose methotrexate

## Abstract

**Simple Summary:**

In this retrospective study, we analyzed the impact of adding high-dose methotrexate to standard chemotherapy on the outcome of patients with diffuse large B-cell lymphoma and skeletal involvement. Our results suggest improved outcome in those who received high-dose methotrexate which should be confirmed in prospective controlled studies.

**Abstract:**

Diffuse large B-cell lymphoma (DLBCL) with extra nodal skeletal involvement is rare. It is currently unclear whether these lymphomas should be treated in the same manner as those without skeletal involvement. We retrospectively analyzed the impact of combining high-dose methotrexate (HD-MTX) with an anthracycline-based regimen and rituximab as first-line treatment in a cohort of 93 patients with DLBCL and skeletal involvement with long follow-up. Fifty patients (54%) received upfront HD-MTX for prophylaxis of CNS recurrence (high IPI score and/or epidural involvement) or because of skeletal involvement. After adjusting for age, ECOG, high LDH levels, and type of skeletal involvement, HD-MTX was associated with an improved PFS and OS (HR: 0.2, 95% CI: 0.1–0.3, *p* < 0.001 and HR: 0.1, 95% CI: 0.04–0.3, *p* < 0.001, respectively). Patients who received HD-MTX had significantly better 5-year PFS and OS (77% vs. 39%, *p* <0.001 and 83 vs. 58%, *p* < 0.001). Radiotherapy was associated with an improved 5-year PFS (74 vs. 48%, *p* = 0.02), whereas 5-year OS was not significantly different (79% vs. 66%, *p* = 0.09). A landmark analysis showed that autologous stem cell transplantation was not associated with improved PFS or OS. The combination of high-dose methotrexate and an anthracycline-based immunochemotherapy is associated with an improved outcome in patients with DLBCL and skeletal involvement and should be confirmed in prospective trials.

## 1. Introduction

Skeletal involvement occurs in 7% to 10% of all lymphomas and includes several clinical entities: localized (LPBL) or multifocal primary bone lymphoma (MPBL) and secondary bone lymphoma (SBL). PBL is defined as one or more lymphoid tumors within bones without supraregional lymph-node involvement or other extra-nodal lesions, whereas SBL is defined as bone involvement of systemic lymphoma [1]. Diffuse Large B Cell Lymphoma (DLBCL) is the most common subtype [2,3]. There is still some debate on whether patients who present with multifocal skeletal disease, concomitant soft tissue, or bone marrow infiltration should be diagnosed as stage IV PBL or SBL [4,5]. Since the advent of positron emission tomography/computed tomography (PET/CT), limited-stage PBL is becoming infrequent, even though this specific entity is the most commonly described in the literature [4]. The clear distinction between these patients is however important, as they have different 5-year overall survival (OS) rates: 84% to 95% in patients with LPBL managed with chemotherapy versus a wide range from 36% to 74% in patients with SBL and MPBL [4,6,7,8]. 

The addition of anti-CD20 immunotherapy (rituximab) to an anthracycline-based regimen has significantly improved the outcome of DLBCL, but its impact on skeletal involvement lymphoma remains unclear [9,10]. High-dose methotrexate (HD-MTX) is currently used for primary central nervous system (PCNS) lymphoma and increasingly, for the prophylaxis of CNS relapse in high-risk patients [11,12]. Also, some reports show that intensified therapy including HD-MTX can benefit some patients with high-risk features such as aaIPI (age-adjusted International Prognostic Index) or double-hit lymphoma, independently of CNS relapse [13,14]. HD-MTX was also used in pediatric patients with lymphoma and skeletal involvement with encouraging results [15]. In another setting, HD-MTX combined with multi-agent chemotherapy led to a dramatic improvement in patients with localized osteosarcoma [16,17,18]. In this study, we evaluated the efficacy of HD-MTX in the treatment of DLBCL with skeletal involvement, in combination with an anthracycline-based chemotherapy regimen and rituximab.

## 2. Materials and Methods

### 2.1. Patients and Setting

All patients with DLBCL and skeletal involvement diagnosed between 2001 and 2014 were retrospectively reviewed from a database of multidisciplinary consultation meetings from the University Hospital Centers of Angers, Rennes, and Tours. Patients were diagnosed by bone or node biopsy based on whichever site was more easily accessible. Pathological confirmation was verified by an experienced pathologist (MCR) according to the World Health Organization (WHO) criteria [1,19]. A CT scan or MRI were performed to assess the type and local extension of bone lesions for all, and PET/CT was performed to assess distant extension in 80/93 patients. In contrast to bone marrow involvement, bone involvement was defined as any lesion within a bone which disrupted bone structure on standard imaging, including Xray, CTscan, and MRI (mostly lytic lesions).

All patients were treated with an anthracycline-based regimen (CHOP or equivalent) in association with rituximab (Appendix A). HD-MTX (1 to 3 g/m^2^) was used with intra-venous hyper hydration and leucovorin rescue, concomitantly to anthracycline-based therapy, in 50/93 patients. HD-MTX was administered as a three-hour infusion on day 2 of chemotherapy. It was mostly used for prophylaxis in patients with high risk for subsequent CNS recurrence (patient with a high IPI score and/or epidural involvement) but some patients received HD-MTX only because of skeletal involvement. At the physician’s discretion, some patients also received platinum-based chemotherapy (DHAP, ESAP), high-dose cytarabine (2g/m^2^ × 2/day), and/or autologous stem cell transplantation (SCT) with a BEAM conditioning regimen as consolidation therapy and/or localized radiotherapy (Appendix A). None of our patients received allogeneic SCT in the first-line setting.

### 2.2. Data Collection

The medical records of the 93 patients identified were reviewed for clinical presentation, radiological findings, response status, and date of last follow-up. We identified three groups: (1) Localized PBL (LPBL); (2) Multifocal PBL (MPBL); and (3) Secondary Bone Lymphoma (SBL). Regional lymph node involvement with or without local extension at the time of diagnosis were considered as LPBL, distant bone marrow involvement as the only other site of extra nodal disease was included as MPBL, and distant lymph node involvement with or without other extra nodal disease was included as SBL [10,20]. The Central Nervous System IPI (CNS-IPI) was calculated as previously described [21]. 

### 2.3. Endpoints

Responses to treatment (overall response rate, ORR, complete response, CR, partial response, PR, stable and progressive disease) and relapse were classified according to the Revised Response Criteria for Malignant lymphoma [22,23]. OS was defined as the interval from diagnosis to death from any cause or last follow-up and progression free survival (PFS) was defined as the interval from diagnosis to disease progression, relapse, or death from any cause.

### 2.4. Statistical Analysis

Continuous variables were presented as median and inter-quartiles and compared using a Mann and Whitney test, and categorical data were presented as numbers and percentages and compared using a Fisher exact test. The median follow-up time was estimated by the reversed Kaplan–Meier method. OS and PFS were estimated according to the Kaplan–Meier method and survival curves were compared using the log-rank test. Survival rates are reported as 5-year PFS and OS +/− standard error. The proportional hazards assumption was confirmed using a graphical approach. Factors associated with OS and PFS were explored by univariable Cox regression analysis. Multivariable analyses were performed using the Cox regression model adjusting for factors included in the International Prognostic Index (IPI) and the type of skeletal involvement. The absence of collinearity between covariables included in the multivariable models was verified. Results were all expressed as hazard ratios (HR) with 95% confidence intervals (95% CI). A landmark analysis was performed to study the impact of autologous SCT that included only patients who were alive and without disease progression or relapse at the median date of autologous SCT, which was 164 days after diagnosis [24]. All tests were two-sided with a significant level *p* < 0.05. No test was used to account for missing values, as these were rare. Statistical analysis was performed with SPSS software version 20 (SPSS Inc., Chicago, IL, USA).

## 3. Results

### 3.1. Patients Characteristics

A total of 93 patients with DLBCL and skeletal involvement were newly diagnosed between 2001 and 2014 and therefore included in the study. Clinical and therapeutic data are summarized in Table 1. Median age was 57 years (48–68) with a male: female ratio of 1.3:1. Most patients had de novo DLBCL, whereas 14 patients (15%) had prior untreated low-grade lymphoma. Most patients (86%) had advanced-stage disease defined by multifocal bone lesion (MPBL, 18%) or disseminated systemic lymphoma with extra nodal disease (SBL, 68%) and 80% of patients had an age-adjusted international prognostic index (aaIPI) score of 2–3. Only 14% of the cohort had localized PBL with 31% presenting with distal bone involvement, the lower limb being the most common involved site. Patients with MPBL and SBL were more likely to have an ECOG > 1 (*p* < 0.001), high LDH levels (*p* < −0.001), a high aaIPI and CNS-IPI (*p* < 0.001 for both comparisons), and epidural involvement (*p* = 0.04).

### 3.2. Outcome of Patients

All 93 patients received anthracycline-based chemotherapy including CHOP (85%) and M-BACOD (13%), while one patient received COPADM and another ACVBP, in combination with rituximab. Response was assessed by PET/CT in 80/93 patients and MRI and CT in the other patients. Initial overall response rate was 98% with only two patients with SBL having stable or progressive disease. The CR rate was 65% (85% for patients with localized PBL, 53% for MPBL, and 64% for SBL) while 33% achieved PR (15% for patients with localized PBL, 47% for MPBL, and 33% for SBL). Among the 31 patients with partial response, 19 (61%) achieved CR at the end of treatment and 4 (13%) remained in PR (Table 2).

First-line consolidation therapy was heterogeneous, with some patients receiving HD-MTX (54%), platinum-based chemotherapy (23%), high-dose cytarabine (40%), and/or autologous stem cell transplantation following BEAM conditioning (42%). Intrathecal chemotherapy was administered to 39 patients (42%) to prevent CNS relapse. Forty patients (43%), mostly with LPBM (62%), received radiotherapy during or after treatment, with a median dose of 30 Grays. The timing and indications of radiotherapy were heterogenous with some patients receiving radiotherapy before chemotherapy for threatening lytic lesions, epidural extension of spine involvement, and/or painful lesions while some received radiotherapy as consolidation.

After a median follow-up of 116 months (69–146), 40 patients (43%) either progressed, relapsed, or died, with a 5-year PFS of 55% +/− 5%. Patients with SBL (51%) were more likely to relapse than patients with either localized PBL (15%) or MPBL (24%). Two patients with localized PBL relapsed, with one experiencing skeletal relapse and one CNS relapse. The sites of relapse included lymph nodes in two patients and skeleton in three patients with MPBL whereas sites of relapse were lymph nodes for 14 patients, skeleton for 13 patients, CNS for 10 patients, and other sites for 3 patients with SBL (Table 2). After a median follow-up of 107 months (64–144), 27 patients (29%) died, with a 5-year OS of 72% +/− 5%.

On univariable analysis, age > 60 years, ECOG > 1, high LDH levels, aaIPI > 1, and type of skeletal involvement (PBL versus SBL) were associated with decreased PFS, whereas age > 60 years and ECOG > 1 were associated with decreased OS (Table 3). All subsequent multivariable analyses were performed adjusting for these factors. PFS and OS were significantly greater in patients with PBL in comparison to patients with SBL (5-year PFS of 83% and 49%, respectively, *p* = 0.004; 5-year OS of 93%, and 64%, respectively, *p* = 0.01) (Figure 1). Both HD-MTX and radiotherapy were associated with improved PFS and OS after univariable analysis (Table 3).

### 3.3. Impact of High Dose Methotrexate

Fifty patients received HD-MTX during initial treatment. These patients were younger (*p* < 0.001), had more advanced-stage disease (*p* = 0.01), had a higher aaIPI (*p* = 0.04) and CNS-IPI (*p* = 0.02), and were more likely to have epidural involvement (*p* < 0.001). They were subsequently more likely to receive platinum-based chemotherapy (*p* = 0.03), HD-cytarabine (*p* < 0.001), and autologous SCT (*p* < 0.001) (Table 4). There were no severe toxicity concerns with HD-MTX, as HD-MTX-related adverse events were mostly transient and reversible in all patients, and usually of minor clinical importance (increased creatinine levels, neutropenia, anemia, increased hepatic enzymes).

After adjusting for age > 60 years, ECOG > 1, high LDH levels, and type of skeletal involvement, HD-MTX remained associated with an improved PFS and OS (HR: 0.2, 95% CI: 0.1–0.3, *p* < 0.001 and HR: 0.1, 95% CI: 0.04–0.3, *p* < 0.001, respectively). Incorporation of other treatments used during consolidation (platinum-based, HD-cytarabine) did not modify this association between HD-MTX and survival. More relapses were observed in patients not receiving HD-MTX (60% versus 24% for those receiving HD-MTX, *p* < 0.001). CNS relapse was not statistically different between the two groups (16% versus 8%, *p* = 0.33). Patients who received HD-MTX had significantly better 5-year PFS and OS (77% vs. 39%, *p* < 0.001 and 83% vs. 58%, *p* < 0.001) (Figure 2). HD-MTX was especially associated with improved survival in patients older than 60 years and with a high CNS-IPI (Appendix A). The impact of HD-MTX in patients with localized and multifocal PBL could not be assessed because of a low number of patients who received HD-MTX in this subgroup. Patients with SBL who received HD-MTX had improved PFS and OS (Appendix A). There was no association between HD-MTX dosing and any parameters.

### 3.4. Impact of Radiotherapy and Autologous Stem Cell Transplantation

Forty patients received radiotherapy before or after initial chemotherapy. There was no significant difference between patients who received or did not recieve radiotherapy, especially regarding the type of skeletal involvement and epidural involvement (Appendix A). After adjusting for age > 60 years, ECOG > 1, high LDH levels, and type of skeletal involvement, radiotherapy remained associated with improved PFS but not OS (HR: 0.5, 95% CI: 0.2–0.9, *p* = 0.03 and HR: 0.5, 95% CI: 0.2–1.2, *p* = 0.13, respectively). Patients who received radiotherapy at any timepoint during treatment had a better 5-year PFS (74% vs. 48%, *p* = 0.02), whereas 5-year OS was not different (79% vs. 66%, *p* = 0.09) (Figure 3).

Because more patients receiving HD-MTX were more likely to subsequently receive autologous SCT following BEAM conditioning, we performed a landmark analysis to study the impact of autologous SCT on the outcome of patients with DLBCL and skeletal involvement. Patients who received autologous SCT had this procedure performed with a median of 164 days after diagnosis. This time point—164 days after diagnosis—was therefore chosen as time zero for the landmark analysis for analyzing patients alive and without progression or relapse. Patients who subsequently received autologous SCT were younger (*p* < 0.001), had more advanced disease (*p* = 0.01), had more epidural involvement (*p* = 0.04), and were more likely to have prior low-grade lymphoma (*p* = 0.002). They were more likely to have received HD-MTX (*p* < 0.001), platinum-based consolidation (*p* = 0.01), and HD-cytarabine (*p* < 0.001) (Appendix A). Autologous SCT had no impact on PFS or OS (Figure 4). After adjusting for autologous SCT, patients who received HD-MTX had better PFS and OS (HR: 0.3, 95% CI: 0.1–0.8, *p* = 0.01 and HR: 0.3, 95% CI: 0.1–0.8, *p* = 0.02, respectively).

## 4. Discussion

In this study, we retrospectively evaluated the impact of HD-MTX as first-line treatment in combination with an anthracycline-based chemotherapy and anti-CD20 immunotherapy in a large series of 93 patients with DLBCL and skeletal involvement with long follow-up. Because HD-MTX has dramatically improved the prognosis of patients with localized osteosarcoma, we tried to clarify its interest in the treatment of lymphoma with skeletal involvement. The adjunction of HD-MTX was significantly associated with an improved outcome with increased 5-year PFS and OS in the 50 patients who received HD-MTX. Radiotherapy was associated with improved PFS but not OS, whereas autologous SCT was not associated with improved outcomes.

Characteristics of patients were representative of lymphoma with skeletal involvement, as previously described [5,7,8]. Most patients (86%) had advanced-stage disease (MPBL, 18%, SBL, 68%). Patients preferentially presented with an axial skeleton involvement with spine and pelvic localizations, especially in MPBL and SBL. An epidural involvement was present in 36% of all patients, which is consistent with other series (7–65%) [7,8,25]. Poor prognostic factors were identical to those previously described: age ≥ 60 years, elevated LDH, altered performance status, high age-adjusted IPI score, and advanced-stage disease [26,27,28]. After multivariable analysis, only altered performance status and SBL were associated with decreased PFS and OS. We confirmed that patients with PBL have a better outcome than patients with SBL with improved 5-year PFS (84%, 82%, and 48% for patients with localized PBL, multifocal PBL, or SBL, respectively, *p* = 0.01) and 5-year OS of 91%, 94%, and 62%, for patients with localized PBL, multifocal PBL, or SBL, respectively, *p* = 0.02) [4,8]. 

In our cohort, patients who received HD-MTX had a better outcome after adjusting for potential confounding factors. HD-MTX has shown anti-lymphoma activity with a CR rate of 77%, especially in CNS lymphoma where it has significantly improved the prognosis [29,30]. Some reports show that intensified therapy including HD-MTX can benefit some patients with high-risk features such as aaIPI or double-hit lymphoma [13,14]. Besides, some authors advocate the significant benefit of dose-intensity of HD-MTX [14,31]. Indeed, HD-MTX enables higher drug concentration in tumor cells [32]. In our study, HD-MTX was associated with an improved outcome in patients with lymphoma and skeletal involvement with greater 5-year PFS and OS (77% vs. 39%, *p* < 0.001 and 83% vs. 58%, *p* < 0.001). HD-MTX was especially associated with improved survival in patients older than 60 years, with a high CNS-IPI, and with SBL.

Whether the positive outcomes observed in our study after adjunction of HD-MTX were due to a decreased CNS relapse rate or a specific anti-lymphoma activity on bone lesions is difficult to settle. There was no statistical difference in CNS relapse rates between patients who received HD-MTX and those who did not (CNS relapse in 8% of patients who received HD-MTX versus 16% in those who did not, *p* = 0.34). Still, our global 12% CNS relapse rate is quite high in comparison to other reports analyzing patients with skeletal involvement (3%) but more in line with previous reports that show that patients with 2 to 3 aaIPI, as it is the case for the majority of our patients, have a CNS relapse risk of 4.2% to 9.7% [8,33]. Although it remains controversial in the rituximab era, bone and epidural involvement are associated with an increased risk of CNS dissemination and HD-MTX is frequently used as first line CNS prophylaxis [12,33,34,35,36,37,38]. Ferreri et al. have recently showed that a HD-MTX-based prophylaxis (3–4 of 3 g/m^2^) can significantly reduce CNS relapses in high-risk patients with specific extra nodal sites (bone, testis, kidney, breast) and/or high aaIPI are considered [39]. Other studies suggest that the addition of HD-MTX to standard immuno-chemotherapy could improve the prognosis of patients, irrespective of the risk of CNS relapse [40]. 

We confirmed the positive role of radiotherapy on PFS in patients with DLBCL and skeletal involvement. It has been previously shown that use of consolidative radiotherapy was associated with an improved outcome in patients with skeletal involvement [8,27,28]. In particular, a large study analyzing 292 patients with skeletal involvement included in prospective trials showed that patients who received radiotherapy had better 3-year event-free survival (75% vs. 36%, *p* < 0.001) [10]. This was mostly observed in patients with advanced disease [8,10,28]. More recent studies suggest that radiotherapy might be omitted in PET-negative patients [41,42]. It is noteworthy that it was difficult to precisely analyze the impact of radiotherapy in our cohort, as some patients received radiotherapy early during treatment and some as consolidation therapy. The positive association of radiotherapy on PFS but not on OS in our cohort, as described previously in some studies, might be due to effective salvage treatments after relapse [8]. 

Our results do not support autologous SCT as first-line treatment for patients with DLBCL and skeletal involvement. We performed a landmark analysis to exclude patients who experienced progression or relapse before or who were lost to follow-up before they could receive autologous SCT, to limit the risk of bias. This analysis shows that patients who received autologous SCT did not have improved PFS or OS. This is line with previous reports that do not show OS improvement in patients with aggressive lymphoma that received first-line autologous SCT [43,44,45]. Nevertheless, one report, including patients who did not all receive rituximab, showed that high-risk patients as evaluated by the IPI could benefit from front-line autologous SCT [46]. This was not confirmed in another high-risk category: patients with double-hit lymphoma [13]. 

Because of its retrospective nature, our study presents some limitations. Patients who received HD-MTX were younger, had more advanced-stage disease, had a higher aaIPI, and were more likely to have epidural involvement. The positive association of HD-MTX with PFS and OS was confirmed after adjustment to these parameters in a multivariable model. In addition, treatments received were heterogenous, with some patients receiving platinum-based consolidation chemotherapy or HD-cytarabine. However, these two consolidation regimens were not associated with improved survival and thus were not included in the multivariable model. 

## 5. Conclusions

The addition of HD-MTX to an anthracycline-based immuno-chemotherapy was associated with an improved outcome in patients with DLBCL and skeletal involvement, especially in advanced-stage disease. Further randomized and prospective trials are required to confirm the efficacy of HD-MTX on DLBCL with skeletal involvement and to determine which subgroup of patients might benefit the most from this treatment.

## Figures and Tables

**Figure 1 cancers-13-02945-f001:**
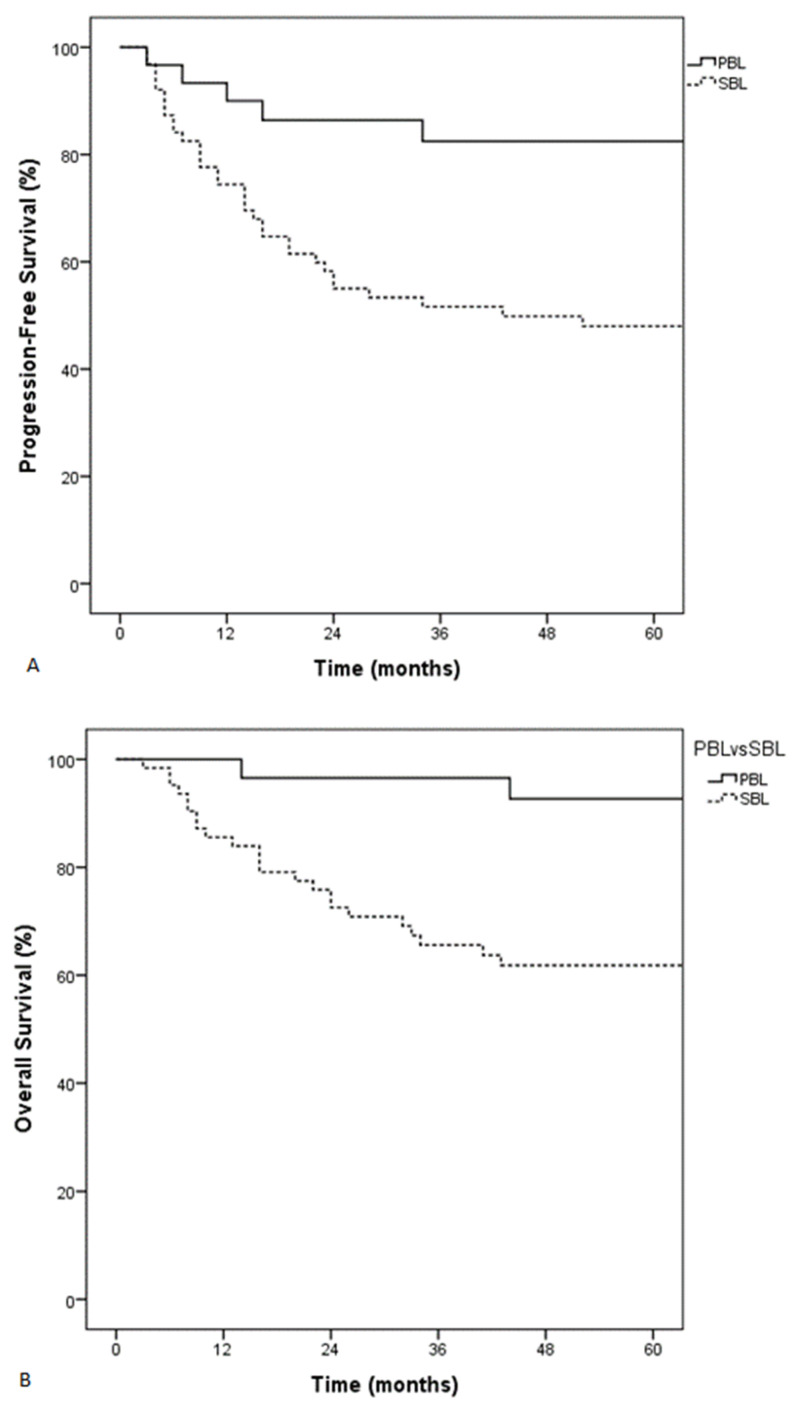
Progression-free survival (PFS) (**A**) and overall survival (OS) (**B**) according to the type of skeletal involvement. PBL, Primary Bone Lymphoma; SBL, Secondary Bone Lymphoma.

**Figure 2 cancers-13-02945-f002:**
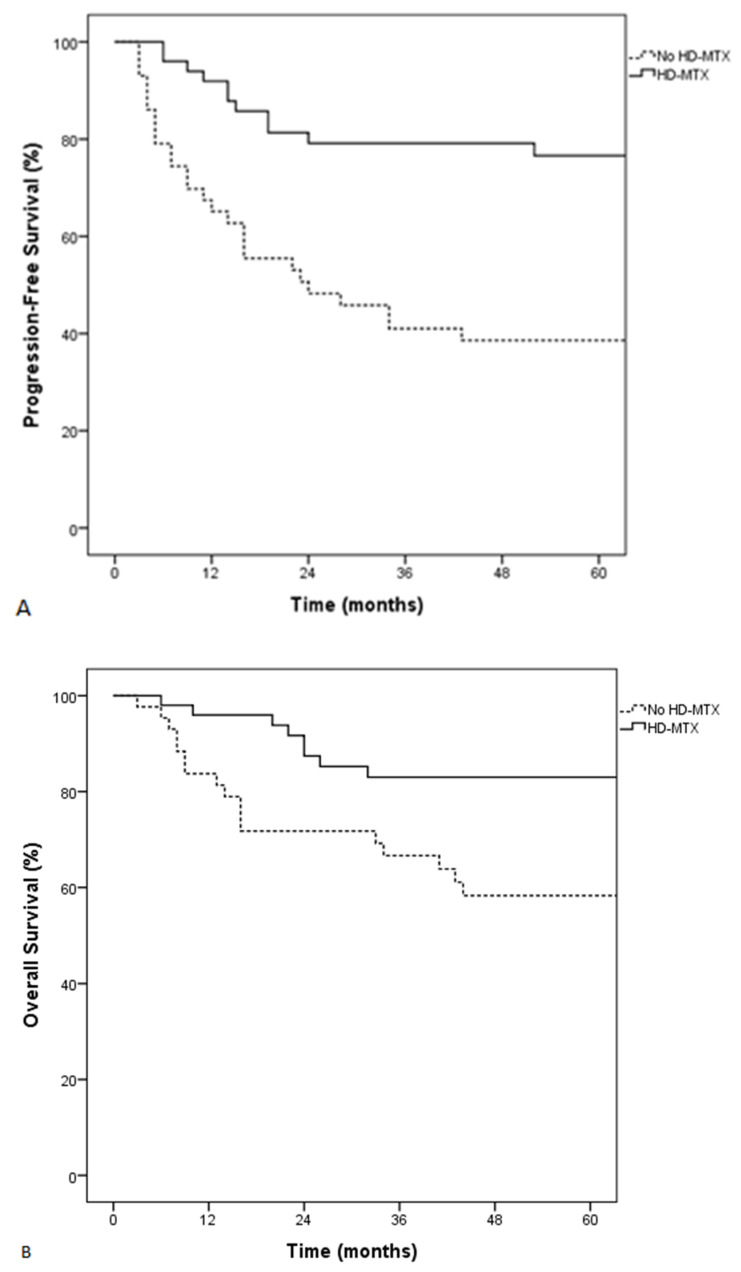
Progression-free survival (PFS) (**A**) and overall survival (OS) (**B**) of patients treated with or without HD-MTX.

**Figure 3 cancers-13-02945-f003:**
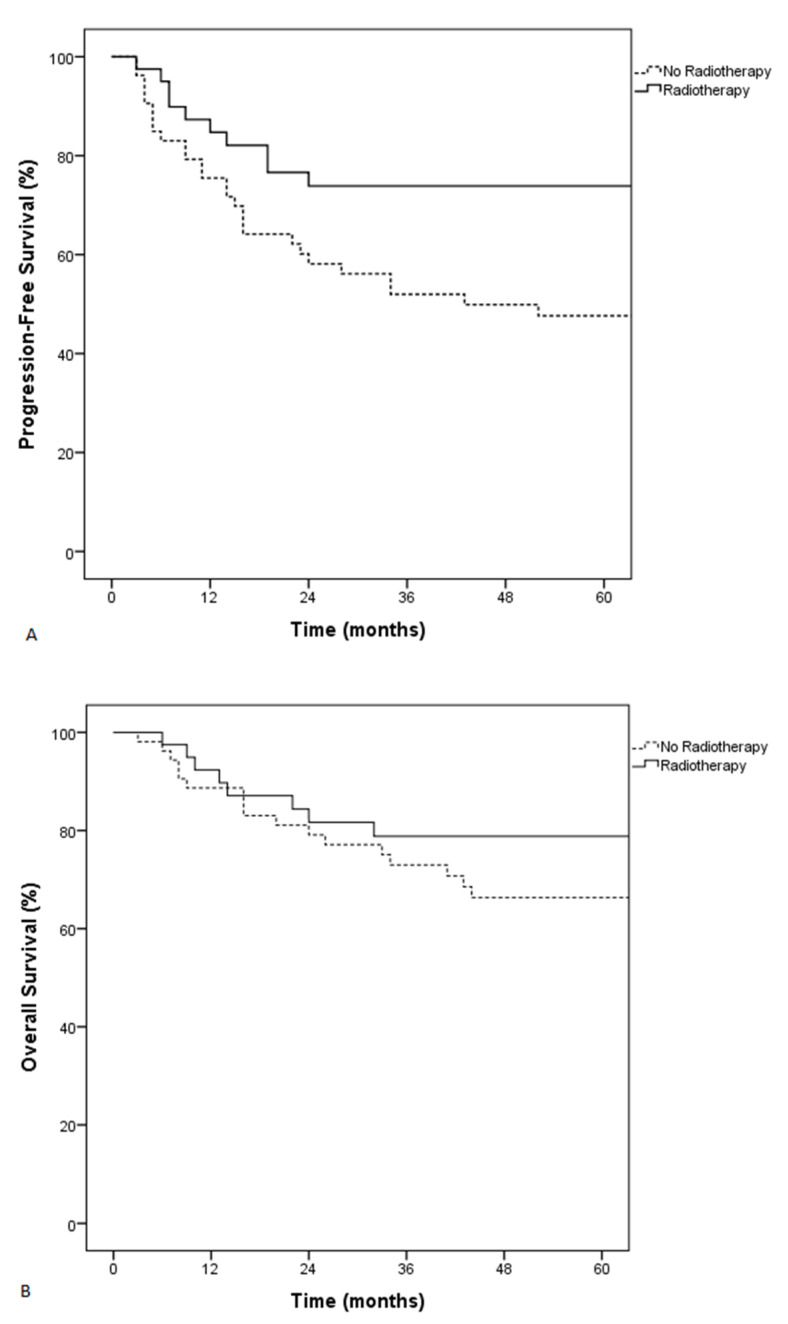
Progression-free survival (PFS) (**A**) and overall survival (OS) (**B**) of patients treated with or without radiotherapy.

**Figure 4 cancers-13-02945-f004:**
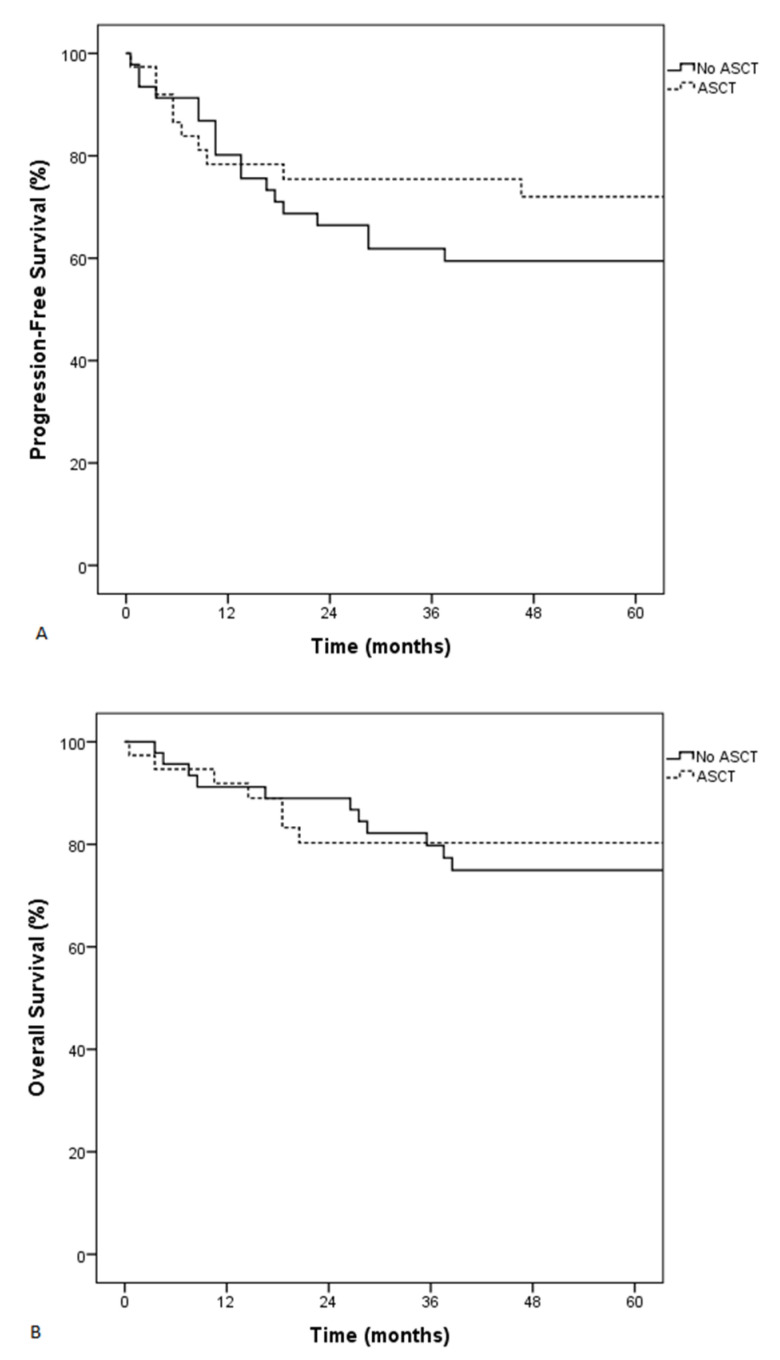
Progression-free survival (PFS) (**A**) and overall survival (OS) (**B**) of patients who did or did not receive autologous stem cell transplantation (ASCT) by landmark analysis.

**Table 1 cancers-13-02945-t001:** Characteristics of the whole cohort and of patients with localized and multifocal primary bone lymphoma and secondary bone lymphoma.

Characteristic	Whole Cohort (*n* = 93)	Localized PBL (*n* = 13)	Multifocal PBL (*n* = 17)	Secondary Bone Lymphoma (*n* = 63)
Age	57 (48–68)	55 (25–63)	51 (45–73)	58 (50–67)
Age > 60 years	42 (45%)	4 (31%)	7 (41%)	31 (49%)
Gender (Male/Female)	52/41	8/5	7/10	37/26
ECOG > 1	46 (50%)	1 (8%)	6 (35%)	39 (62%)
High LDH serum levels	71 (76%)	4 (31%)	14 (82%)	53 (84%)
Stage III/IV	80 (86%)	0	17 (100%)	63 (100%)
aaIPI				
0–1	19 (20%)	12 (92%)	2 (12%)	5 (8%)
2–3	74 (80%)	1 (8%)	15 (88%)	58 (92%)
CNS-IPI				
0–1	13 (14%)	10 (77%)	1 (6%)	2 (3%)
2–3	33 (35%)	3 (23%)	10 (59%)	20 (32%)
≥4	47 (51%)	0	6 (35%)	41 (65%)
Sites of Skeletal Involvement				
Axial	84 (90%)	9 (69%)	15 (88%)	60 (95%)
Skull	7 (8%)	1 (8%)	1 (6%)	5 (8%)
Vertebral	58 (62%)	5 (39%)	10 (59%)	43 (68%)
Rib Cage	23 (25%)	1 (8%)	5 (29%)	17 (27%)
Pelvis	42 (45%)	2 (15%)	9 (53%)	31 (49%)
Distal	36 (39%)	4 (31%)	7 (41%)	25 (40%)
Lower Limb	23 (25%)	3 (23%)	2 (12%)	18 (29%)
Upper Limb	21 (23%)	1 (8%)	7 (41%)	13 (21%)
Epidural Involvement	33 (36%)	2 (15%)	10 (59%)	21 (33%)
Bone Marrow Involvement	36 (39%)	1 (8%)	3 (18%)	32 (51%)
Prior Low-Grade Lymphoma	14 (15%)	0	1 (6%)	13 (21%)
Treatment				
Initial Chemotherapy Regimen				
CHOP	79 (85%)	11 (85%)	16 (94%)	52 (83%)
MBACOD	12 (13%)	2 (15%)	1 (6%)	9 (14%)
Other	2 (2%)	0	0	2 (3%)
High-Dose Methotrexate	50 (54%)	2 (15%)	11 (65%)	37 (59%)
Platinum-based Consolidation	21 (23%)	3 (23%)	2 (12%)	16 (25%)
High-Dose Cytarabine Consolidation	37 (40%)	4 (31%)	7 (41%)	26 (41%)
Intra-Thecal Therapy	39 (42%)	4 (31%)	12 (71%)	23 (37%)
Radiotherapy	40 (43%)	8 (62%)	8 (47%)	24 (38%)
Autologous SCT	39 (42%)	1 (8%)	9 (53%)	29 (46%)

aaIPI, age-adjusted International Prognostic Index; CNS-IPI, Central Nervous System IPI; ECOG, Eastern Cooperative Oncology Group; LDH: Lactate DeHydrogenase; PBL, Primary Bone Lymphoma; SCT, Stem Cell Transplantation.

**Table 2 cancers-13-02945-t002:** Outcome of patients.

	Whole Cohort (*n* = 93)	Localized PBL (*n* = 13)	Multifocal PBL (*n* = 17)	Secondary Bone Lymphoma (*n* = 63)
Initial Response				
Complete Remission	60 (65%)	11 (85%)	9 (53%)	40 (64%)
Partial Response	31 (33%)	2 (15%)	8 (47%)	21 (33%)
Stable/Progressive Disease	2 (2%)	0	0	2 (3%)
Best Response				
Complete Remission	66 (71%)	13 (100%)	13 (76%)	40 (64%)
Partial Response	22 (24%)	0	1 (6%)	21 (3%)
Relapse	38 (41%)	2 (15%)	4 (24%)	32 (51%)
Lymph Node	16 (17%)	0	2 (12%)	14 (22%)
Skeletal	17 (18%)	1 (8%)	3 (18%)	13 (21%)
CNS	11 (12%)	1 (8%)	0	10 (16%)
Other	3 (3%)	0	0	3 (5%)

CNS, Central Nervous System; PBL, Primary Bone Lymphoma.

**Table 3 cancers-13-02945-t003:** Univariable analysis of factors associated with PFS and OS.

	PFS		OS	
Characteristic	HR (95% CI)	*p*	HR (95% CI)	*p*
Age > 60 years	2.8 (1.5–5.2)	0.002	2.4 (1.1–5)	0.02
Gender (M/F)	0.99 (0.5–1.8)	0.97	0.8 (0.4–1.6)	0.49
ECOG > 1	3.1 (1.6–5.9)	0.001	4.4 (1.9–9.9)	<0.001
High LDH serum levels	3.1 (1.2–7.8)	0.02	2.4 (0.9–7)	0.1
Stage III/IV	3.9 (0.9–16)	0.06	5.3 (0.7–39)	0.1
aaIPI				
0–1	Ref.		Ref.	
2–3	4.4 (1.3–14)	0.01	2.6 (0.8–8.7)	0.11
CNS-IPI				
0–1	Ref.		Ref.	
2–3	2.2 (0.5–10)	0.31	0.99 (0.19–5.1)	0.99
≥4	6.4 (1.5–27)	0.01	4.3 (1–19)	0.05
Type of Lymphoma				
Localized PBL	Ref.		Ref.	
Multifocal PBL	1.8 (0.4–9.5)	0.47	2.1 (0.2–20)	0.52
Secondary Bone Lymphoma	4.6 (1.1–19)	0.04	6.4 (0.9–47)	0.07
Sites of Skeletal Involvement				
Axial	3 (0.7–12)	0.14	3.8 (0.5–28)	0.19
Distal	0.9 (0.5–1.7)	0.78	1.1 (0.5–2.2)	0.85
Epidural Involvement	0.6 (0.3–1.2)	0.15	0.5 (0.2–1.2)	0.13
Bone Marrow Involvement	1.4 (0.8–2.6)	0.26	1.4 (0.7–2.9)	0.34
Prior Low-Grade Lymphoma	0.99 (0.4–2.4)	0.99	1.3 (0.5–3.5)	0.56
Treatment				
Initial Chemotherapy Regimen				
CHOP	Ref.		Ref.	
MBACOD	0.7 (0.3–1.7)	0.38	0.8 (0.3–2.4)	0.73
Other	0.9 (0.1–6.3)	0.88	1.5 (0.2–11)	0.7
High-Dose Methotrexate	0.3 (0.1–0.5)	<0.001	0.2 (0.1–0.5)	0.001
Platinum-based Consolidation	1.1 (0.5–2.2)	0.82	1.2 (0.5–2.6)	0.7
High-Dose Cytarabine Consolidation	0.6 (0.3–1.2)	0.14	0.7 (0.3–1.5)	0.35
Intra-Thecal Therapy	0.8 (0.4–1.4)	0.38	0.8 (0.4–1.6)	0.48
Radiotherapy	0.5 (0.3–0.9)	0.02	0.5 (0.2–1.1)	0.01

aaIPI, age-adjusted International Prognostic Index; CNS-IPI, Central Nervous System IPI; ECOG, Eastern Cooperative Oncology Group; LDH: Lactate DeHydrogenase; PBL, Primary Bone Lymphoma.

**Table 4 cancers-13-02945-t004:** Characteristics of the whole cohort and of patients who did or did not receive HD-MTX.

Characteristic	Whole Cohort (*n* = 93)	No HD-MTX (*n* = 43)	HD-MTX (*n* = 50)
Age	57 (48–68)	63 (54–72)	52 (44–60)
Age > 60 years	42 (45%)	29 (67%)	13 (26%)
Gender (M/F)	52/41	23/20	29/21
ECOG > 1	46 (50%)	20 (47%)	26 (52%)
High LDH serum levels	71 (76%)	33 (77%)	38 (76%)
Stage III/IV	80 (86%)	32 (74%)	48 (96%)
aaIPI			
0–1	19 (20%)	13 (30%)	6 (12%)
2–3	74 (80%)	30 (70%)	44 (88%)
CNS-IPI			
0–1	13 (14%)	9 (21%)	4 (8%)
2–3	33 (35%)	9 (21%)	24 (48%)
≥4	47 (51%)	25 (58%)	22 (44%)
Type of Lymphoma			
Localized PBL	13 (14%)	11 (26%)	2 (4%)
Multifocal PBL	17 (18%)	6 (14%)	11 (22%)
Secondary Bone Lymphoma	63 (68%)	26 (60%)	37 (74%)
Sites of Skeletal Involvement			
Axial	84 (90%)	37 (86%)	47 (94%)
Skull	7 (8%)	3 (7%)	4 (8%)
Vertebral	58 (62%)	24 (56%)	34 (68%)
Rib Cage	23 (25%)	9 (21%)	14 (28%)
Pelvis	42 (45%)	21 (49%)	21 (42%)
Distal	36 (39%)	17 (40%)	19 (38%)
Lower Limb	23 (25%)	11 (26%)	12 (24%)
Upper Limb	21 (23%)	10 (23%)	11 (22%)
Epidural Involvement	33 (36%)	6 (14%)	27 (54%)
Bone Marrow Involvement	36 (39%)	15 (35%)	21 (42%)
Prior Low-Grade Lymphoma	14 (15%)	2 (5%)	12 (24%)
Treatment			
Initial Chemotherapy Regimen			
CHOP	79 (85%)	43 (100%)	36 (72%)
MBACOD	12 (13%)	0	12 (24%)
Other	2 (2%)	0	2 (4%)
Platinum-based Consolidation	21 (23%)	5 (12%)	16 (32%)
High-Dose Cytarabine Consolidation	37 (40%)	5 (12%)	32 (64%)
Intra-Thecal Therapy	39 (42%)	14 (33%)	25 (50%)
Radiotherapy	40 (43%)	14 (33%)	26 (52%)
Autologous SCT	39 (42%)	6 (14%)	33 (66%)

aaIPI, age-adjusted International Prognostic Index; CNS-IPI, Central Nervous System IPI; ECOG, Eastern Cooperative Oncology Group; LDH: Lactate DeHydrogenase; PBL, Primary Bone Lymphoma; SCT, Stem Cell Transplantation.

## Data Availability

The data presented in this study are available on request from the corresponding author.

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
