# Peer review of "Impact of High-Dose Methotrexate on the Outcome of Patients with Diffuse Large B-Cell Lymphoma and Skeletal Involvement"

_cancers, 2021, doi:10.3390/cancers13122945_

Round 1

Reviewer 1 Report

This is a retrospective study about the role of high-dose MTX in conjunction with rituximab-CHOP or other regimens in DLBCL patients with skeletal involvement. Although this study analyzed relatively large number of patients, several limitations are found as follows.

First, the dosage and schedule of high-dose MTX were not clearly described. Furthermore, the induction treatments were too heterogeneous. The information about the treatment should be more clarified, and more detailed data should be provided.

Second, what is the rationale for the use of MTX in DLBCL patients with skeletal involvement? Did you use it for CNS prophylaxis? But, skeletal involvement itself is not sufficient for CNS prophylaxis. Are there any evidence supproting the efficacy of MTX in DLBCL involving bones?

Third, how could you discriminate bone involvement from bone marrow involvement? Does skeletal involvement mean the involvement of bone cortex with tumor cells?

Author Response

RESPONSE TO REVIEWER #1

  1. First, the dosage and schedule of high-dose MTX were not clearly described. Furthermore, the induction treatments were too heterogeneous. The information about the treatment should be more clarified, and more detailed data should be provided.

We agree that dosage and schedule for HD-MTX should be more precisely described. Our patients received a three-hour infusion of 1 to 3 mg/m2 on day 2 of each treatment cycle. This has been added in the “Methods” section, page 2, which now reads: “HD-MTX was administered as a three-hour infusion on day 2 of chemotherapy.

  1. Second, what is the rationale for the use of MTX in DLBCL patients with skeletal involvement? Did you use it for CNS prophylaxis? But, skeletal involvement itself is not sufficient for CNS prophylaxis. Are there any evidence supporting the efficacy of MTX in DLBCL involving bones?

Indeed, we should describe in more details our rationale for using HD-MTX in treating patients with HD-MTX. Other than its potential use in CNS prophylaxis, we were interested that it was used with good results in the treatment of osteosarcoma, another bone cancer. Also, we came across a study in pediatric patients with lymhoma and skeletal involvement who seemed to have outstanding results with HD-MTX. The rationale for using HD-MTX is now described in more details in the “Introduction” section, page 2, which now reads: “In another setting, HD-MTX combined with multi-agent chemotherapy led to a dramatic improvement in patients with localized osteosarcoma. HD-MTX was also used in pediatric patients with lymphoma and skeletal involvement with encouraging results.” The rationale for HD-MTX was also recalled in the “Discussion” section, page 11, which now reads: “Because HD-MTX has dramatically improved the prognosis of patients with localized osteosarcoma, we tried to clarify its interest in the treatment of lymphoma with skeletal involvement.

  1. Third, how could you discriminate bone involvement from bone marrow involvement? Does skeletal involvement mean the involvement of bone cortex with tumor cells?

We agree that we should better define bone involvement as it is sometimes difficult to differentiate if from bone marrow involvement. We used a radiological definition where bone involvement was considered when there was morphological disruption of the skeletal structure on imaging whereas bone marrow involvement was considered if bone marrow biopsy was positive and/or PET-scan showed hypermetabolism without any lesion visualized on standard imaging. This has been clarified in the “Methods” section, page 2, which now reads: “In contrast to bone marrow involvement, bone involvement was defined as any lesion within bone which disrupted bone structure on standard imaging, including Xray, CTscan, and MRI (mostly lytic lesions).

Reviewer 2 Report

This is an interesting manuscript that retrospectively analyzes 93 DLBCL patients with bone involvement treated over 2 decades in collaborating lymphoma treatment centers.

The major goal of the study is to analyze the effect of high-dose methotrexate therapy in this relatively rare disease condition. As this is a not preplanned retrospective analysis, a severe bias by the treating physician is introduced: patients treated with high-dose MTX are clearly different in every important aspect of their condition from those who were not receiving this therapy. Nevertheless, due to the multicentric nature of the study and proper statistical analysis, the conclusions of the paper have merit.

Since the study is retrospective and the patients were not uniformly treated, the question still emerges about the specifics of the MTX therapy that is central to the paper. 1-3 g/m2 MTX dose is given. However, there is a clear difference in modalities whether this MTX dose was administered in rapid infusion or rather as 24h continuous iv infusion. Data about this treatment should be included in the tables and in the result section - if sufficient data are available for subgroup forming and evaluation than that should be done.

Some clarification is also needed about the distiction of an intraosseal versus paramedullary growth of lymhoma associated with the bone. Additionally, clarification is needed about the distinction of bone versus bone marrow involvement.

Author Response

RESPONSE TO REVIEWER #2

  1. Since the study is retrospective and the patients were not uniformly treated, the question still emerges about the specifics of the MTX therapy that is central to the paper. 1-3 g/m2 MTX dose is given. However, there is a clear difference in modalities whether this MTX dose was administered in rapid infusion or rather as 24h continuous iv infusion. Data about this treatment should be included in the tables and in the result section - if sufficient data are available for subgroup forming and evaluation than that should be done.

We agree that dosage and schedule for HD-MTX should be more precisely described. Our patients received a three-hour infusion of 1 to 3 mg/m2 on day 2 of each treatment cycle. This has been added in the “Methods” section, page 2, which now reads: “HD-MTX was administered as a three-hour infusion on day 2 of chemotherapy.

  1. Some clarification is also needed about the distinction of an intraosseal versus paramedullary growth of lymphoma associated with the bone. Additionally, clarification is needed about the distinction of bone versus bone marrow involvement.

We agree that we should better define bone involvement as it is sometimes difficult to differentiate if from bone marrow involvement. We used a radiological definition where bone involvement was considered when there was morphological disruption of the skeletal structure on imaging whereas bone marrow involvement was considered if bone marrow biopsy was positive and/or PET-scan showed hypermetabolism without any lesion visualized on standard imaging. This has been clarified in the “Methods” section, page 2, which now reads: “In contrast to bone marrow involvement, bone involvement was defined as any lesion within bone which disrupted bone structure on standard imaging, including Xray, CTscan, and MRI (mostly lytic lesions).

Reviewer 3 Report

Mercier M et al retrospectively evaluated the impact of HD-MTX as first-line treatment in combination with an anthracycline-based chemotherapy and anti-CD20 immunotherapy in a large series of 93 patients with DLBCL.

General Comment. Authors presented with a well-written manuscript a therapeutic experience of a rare condition. Unfortunately, this study has two limitations: 1-) it is a retrospective analyses; 2-) it is about a biologically different entities: primary bone lymphoma is a biologically different disease comparing with secondary bone lymphoma. The latter is a sort of metastatic disease which needs to be treated with a more intensive therapy. 

Major Issues. HD-MTX is a cornerstone of acute lymphoblastic leukemia. Thus this study confirmed that increasing intensity with HDMTX improved outcome. Even the application of autologous SCT, benefits from HD-MTX. It is well known that the success of SCT procedure is strictly related to the remission status, thus to the treatment adopted before SCT.  

However, it is cited only the autologous transplantation. Did any of these cases receive an allogeneic SCT, which is more effective than autologous to eradicate aggressive DLBCL as well as Lymphoblastic Lymphoma? Would authors please insert a comment on this therapeutic opportunity, since in this study the average age was 42?  

Minor issue

row 300. Please insert the reference regarding the results of autologous SCT as first-line therapy, showing no OS improvement in patients with aggressive lymphoma.

Author Response

  1. HD-MTX is a cornerstone of acute lymphoblastic leukemia. Thus, this study confirmed that increasing intensity with HDMTX improved outcome. Even the application of autologous SCT, benefits from HD-MTX. It is well known that the success of SCT procedure is strictly related to the remission status, thus to the treatment adopted before SCT.

However, it is cited only the autologous transplantation. Did any of these cases receive an allogeneic SCT, which is more effective than autologous to eradicate aggressive DLBCL as well as Lymphoblastic Lymphoma? Would authors please insert a comment on this therapeutic opportunity, since in this study the average age was 42?

We agree that we should give precise details regarding which treatment were received by each patient. It was our policy to consider autologous SCT, but not allogeneic SCT, for some of our patients with lymphoma and skeletal as we were in the first-line setting. None of our patients therefore received allogeneic SCT in our cohort. This has been clarified in the “Methods” section, page 2, which now reads: “None of our patients received allogeneic SCT in the first-line setting.

  1. row 300. Please insert the reference regarding the results of autologous SCT as first-line therapy, showing no OS improvement in patients with aggressive lymphoma.

References were inserted as requested.

Round 2

Reviewer 1 Report

In my opinion, this study still does not reach the quality that can be acceptable for publication. The current message of this study reminas unclear to readers.

Reviewer 2 Report

Improved manuscipt with better description of the data. Still, a retrospective but interesting analysis.